# Cardiac MRI for COVID-19-Related Late Myocarditis: Functional Parameters and T1 and T2 Mapping

**DOI:** 10.3390/diagnostics15192441

**Published:** 2025-09-25

**Authors:** Sena Ünal, Elif Peker, Sena Bozer Uludağ, Sezer Nil Yılmazer Zorlu, Ruhi Erdem Ergüden, Arda Ayhan Hekimoğlu

**Affiliations:** Faculty of Medicine, Ankara University, Ankara 06610, Turkey; drsenaunal@gmail.com (S.Ü.); senabozer@gmail.com (S.B.U.); sezernilyilmazer@gmail.com (S.N.Y.Z.); erdemerguden@hotmail.com (R.E.E.); dr.ardahekimoglu@gmail.com (A.A.H.)

**Keywords:** COVID-19 myocarditis, LGE, myocarditis, T1 mapping, T2 mapping

## Abstract

**Background/Objectives**: Myocarditis is a recognized complication of COVID-19 infection, with potential long-term cardiac sequelae. While acute cardiac involvement has been frequently reported, late-stage myocardial effects remain less well characterized. Cardiac magnetic resonance (CMR) imaging, particularly T1 and T2 mapping, offers non-invasive tissue characterization for myocardial inflammation and fibrosis. This study aimed to evaluate segmental myocardial tissue changes in patients with late-stage COVID-19–related myocarditis using CMR and compare findings with patients with non-COVID-19 myocarditis and healthy controls. **Methods**: This retrospective, single-center study included 25 patients with clinically suspected COVID-19 myocarditis who underwent CMR between 36 and 565 days post-infection. T1 and T2 mapping values and late gadolinium enhancement (LGE) patterns were assessed and compared with 14 non-COVID-19 myocarditis patients and 19 healthy controls. Subgroup analyses were performed according to vaccination status and left ventricular ejection fraction (LVEF). **Results**: Patients with reduced LVEF had significantly higher T1 and T2 values in several myocardial segments. Compared to controls, the COVID-19 myocarditis group showed significantly elevated T1 values in all segments except 2 and 3. No significant difference in T2 values was observed. LGE was present in 61% of COVID-19 myocarditis patients, predominantly with a subepicardial pattern. No significant differences were observed between vaccinated and unvaccinated patients. **Conclusions**: Late-stage COVID-19 myocarditis is associated with persistent segmental myocardial tissue abnormalities, particularly elevated T1 values and subepicardial LGE. Segmental CMR mapping may provide additional diagnostic value in identifying residual myocardial injury in patients with ongoing cardiac symptoms after COVID-19 infection.

## 1. Introduction

The Coronavirus Disease 2019 (COVID-19) pandemic has caused widespread morbidity and mortality with multisystem involvement. While the acute manifestations of the infection are well-documented, less is known about its long-term effects. In addition to respiratory complications, COVID-19 can affect multiple organ systems, including the cardiovascular system [1]. Similar to other viral infections, COVID-19 has the potential to cause myocardial injury and myocarditis.

Cardiac magnetic resonance imaging (MRI) is a state-of-the-art, non-invasive imaging modality that plays a central role in the diagnosis of myocarditis. It enables both morphological and functional assessment of the myocardium. Beyond conventional MRI sequences, T1 and T2 mapping techniques have emerged as promising tools for detailed myocardial tissue characterization [2].

COVID-19–related myocarditis is one of several cardiac complications associated with the infection, alongside ischemic injury and arrhythmias [3]. Even if not evaluated during the acute phase, myocardial injury due to myocarditis may lead to long-term cardiac dysfunction, highlighting the need for follow-up imaging in the post-acute period [4]. Cardiac MRI facilitates the detection of scar tissue via late gadolinium enhancement (LGE) imaging, while T1 and T2 mapping provide additional insights into myocardial fibrosis and edema, respectively [5].

Although numerous studies have highlighted the role of cardiac MRI in the assessment of COVID-19–related myocarditis, to our knowledge, no prior research has provided a segmental analysis of T1 and T2 mapping or directly compared COVID-19 myocarditis with pre-pandemic myocarditis.

The aim of this study was to evaluate cardiac functional parameters, the presence and pattern of contrast enhancement, and segmental T1 and T2 mapping values in patients with COVID-19–related myocarditis. These findings were compared to those of patients with non-COVID-19 myocarditis and individuals without known cardiac pathology.

## 2. Materials and Methods

This retrospective, single-center, case–control study was approved by the Institutional Review Board of Ankara University, Faculty of Medicine, (Approval code: 2023000659. Approval date: 13 November 2023), and written informed consent was obtained from all participants.

Cardiac magnetic resonance (CMR) images of 45 patients with suspected myocarditis who underwent CMR between March 2020 and December 2022 during the COVID-19 pandemic were retrospectively reviewed. The following patients were excluded: those with symptoms suggestive of myocarditis prior to COVID-19 infection (n = 2), post-COVID-19 vaccination (n = 4), with non-COVID-19 myocarditis (n = 2), a previous history of myocarditis (n = 1), coexisting cardiac disease (n = 2), malignancy (n = 1), poor image quality (n = 2), missing clinical data (n = 2), or absence of basal segment T1 or T2 maps (n = 4).

The final COVID-19 myocarditis group (n = 25) included patients with the following: [1] a confirmed history of COVID-19 infection via PCR, ref. [2] cardiac symptoms such as chest pain or dyspnea, ref. [3] elevated cardiac biomarkers during the acute phase, and [4] supportive ECG findings. The time interval between the PCR test and the CMR examination ranged from 36 to 565 days (mean: 164 ± 148.5; median: 115 days). To address potential heterogeneity related to timing, patients were stratified into early (≤90 days) and late (>90 days) post-infection subgroups based on the interval between positive PCR and CMR imaging. Clinical records were reviewed for symptoms such as dyspnea, chest pain, palpitations, and reduced exercise tolerance. Symptom persistence beyond 4 weeks was considered a ‘prolonged clinical course’.

Since the COVID-19 myocarditis group predominantly consisted of patients in the late post-infection period, we selected non-COVID myocarditis cases who also underwent late-phase imaging for a more accurate comparison. In the non-COVID myocarditis group, the interval between diagnosis and CMR ranged from 22 to 1815 days (mean: 319 ± 597 days; median: 60 days). Although the non-COVID group underwent MRI at a numerically later time point compared to the COVID-19 group (319 vs. 164.4 days), this difference was not statistically significant.

A non-COVID-19 myocarditis group (n = 27) and a healthy control group (n = 19) were formed using previously published patient data [6]. The non-COVID-19 myocarditis group consisted of patients who underwent CMR between July 2016 and May 2019, within ≤7 days of symptom onset, and had typical CMR findings for acute myocarditis. The control group included subjects with no known systemic inflammatory or chronic diseases, no long-term use of anti-inflammatory medications, and normal cardiac imaging findings.

In total, 58 patients’ CMR images were independently analyzed by two radiologists with 8 and 3 years of experience, respectively.

All imaging was performed using a 1.5T scanner (Aera, Siemens Healthineers, Erlangen, Germany) with an 18-channel body coil. The protocol was based on recommendations of the European Society of Cardiology and the Society for Cardiovascular Magnetic Resonance [7,8]. Cine images in short-axis (SAX) and 2-, 3-, and 4-chamber views were obtained using a balanced steady-state free precession (bSSFP) sequence during breath-hold and with retrospective cardiac gating (spatial resolution: 1.63 × 1.63 × 8 mm; flip angle: 80°; TR/TE: 42.98/1.33 ms).

T1 mapping was performed using a modified Look-Locker inversion recovery (MOLLI 5[3]3) sequence and T2 mapping using a single-shot True-FISP readout sequence. All mapping sequences were vendor-supplied (Siemens, Erlangen, Germany). Basal short-axis slices were standardized using the “5 into 3” method as described previously [9]. Late gadolinium enhancement (LGE) images were acquired approximately 9–10 min after intravenous injection of 0.1 mmol/kg gadobutrol, with inversion time adjusted based on Look-Locker scout images.

CMR analysis was performed using a dedicated workstation (syngo.via, Siemens Healthineers). Left ventricular volumes were calculated from cine images by manually tracing endocardial borders in end-diastole and end-systole. End-diastolic volume (EDV) and end-systolic volume (ESV) were calculated using Simpson’s rule. LGE presence and pattern were assessed in 23 of 25 COVID-19 myocarditis patients.

T1 and T2 mapping measurements were performed on the basal short-axis slice, segmented into six standard myocardial segments as previously described [10], ensuring exclusion of the blood pool (Figure 1).

T1 and T2 values in the COVID-19 myocarditis group were compared with those of the non-COVID-19 myocarditis and control groups. Subgroup analyses were also performed based on ejection fraction (EF) and vaccination status. Mapping values exceeding two standard deviations (SD) above the control mean were defined as elevated.

All statistical analyses were performed using SPSS software 15.0 (IBM Corp., Armonk, NY, USA). Descriptive statistics for continuous variables are presented as mean ± SD. The Shapiro–Wilk test was used to assess normality. For comparisons of two related samples, the paired Student’s *t*-test was used if parametric assumptions were met; otherwise, the Mann–Whitney U test was applied. Spearman’s rho was used for correlation analyses. Hedges’ g was used to quantify the effect size between vaccinated and unvaccinated groups due to the small and unequal sample sizes. Additionally, post hoc power analyses (α = 0.05) were conducted to assess the statistical power of each comparison. A two-sided *p*-value < 0.05 was considered statistically significant.

## 3. Results

The mean age of patients in the COVID-19 myocarditis group was 46 ± 19 years. Of these, 20% (n = 5) had received at least one dose of a COVID-19 vaccine.

The most common presenting symptoms were palpitations (n = 12), chest pain (n = 9) and dyspnea (n = 6), followed by syncope (n = 1) and reduced exercise tolerance (n = 1).

Persistent symptoms beyond 4 weeks were reported in 78% (n = 14/18) of patients with available medical records, most commonly palpitations (n = 11), chest pain (n = 2) and dyspnea (n = 2). Among the cohort, four patients experienced major adverse cardiac outcomes, which included new-onset heart failure, catheter ablation for arrhythmia, ICD implantation, and a non-fatal myocardial infarction. When patients with and without persistent symptoms (based on available medical records) were compared, no significant differences were observed in myocardial mapping values, except for the T1 value obtained from the inferior wall, which was significantly higher in the symptomatic group (1049.7 ms vs. 984.7 ms, *p* = 0.024).

Morphological and functional assessment revealed arrhythmia in 5 patients, wall motion abnormalities in 8 patients, and pericardial effusion in 12 patients. Left ventricular ejection fraction (EF) was reduced (<50%) in 28% of patients (n = 7), with a minimum EF of 24% and a maximum of 69% (mean EF: 54%).

The highest segmental native T1 values were observed in segment 4 (1035 ± 59.2 ms), and the highest T2 values were noted in segments 4 and 5 (48.7 ± 4.5 ms and 48.7 ± 4.9 ms, respectively). Increased T1 values (defined as >2 SD above the reference) were found in 5 patients (20%). Segment 6 demonstrated elevated T1 values in all five patients, and segment 5 in three of them.

Subgroup Analyses: Vaccinated and Unvaccinated

No statistically significant differences were observed between vaccinated and unvaccinated patients in terms of EDV, ESV, EF, or T1 and T2 mapping values (*p* > 0.05 for all comparisons). Effect size analysis revealed small Hedges’ g values for all myocardial segments for both T1 and T2 mapping, with the exception of segment 2 T2 (g = 0.62), indicating a moderate effect. However, due to the small sample size in the unvaccinated subgroup (n = 5), post hoc power remained low (<0.25) for all comparisons.

Subgroup analysis according to EF:

When patients were stratified by EF into low (<50%) and high (≥50%) subgroups, several statistically significant differences were identified. Patients with low EF had significantly higher:(a)Segment 5 T1 values (*p* = 0.025)(b)Segment 5 T2 values (*p* = 0.020)(c)Segment 2 T1 values (*p* = 0.023)(d)Segment 6 T2 values (*p* = 0.044) (Table 1)

Moderate negative correlations were observed between EF and T1 values in segments 1, 2, 3, and 5, and between EF and T2 values in segment 6 (Table 2).

Group Comparisons: COVID-19 Myocarditis vs. Early and Late Non-COVID-19 Myocarditis

T1 and T2 mapping values were significantly higher in the non-COVID-19 myocarditis group compared to the COVID-19 myocarditis group in segments 1, 4, 5, and 6. No significant differences were noted in segments 2 and 3 (Table 3). No statistically significant differences were observed in segmental T1 and T2 mapping values between patients with COVID-19 myocarditis and those with late-stage non-COVID-19 myocarditis.

Group Comparisons: Early (≤90 days) and Late (>90 days) COVID Myocarditis vs. Controls

Although patients were stratified into early (≤90 days post-COVID) and late (>90 days post-COVID) groups based on the interval between infection and MRI, no statistically significant differences were observed between the groups in terms of segmental T1 and T2 values (*p* > 0.05 for all comparisons). No statistically significant differences in T1 or T2 mapping values were observed between the control group and either the early or late COVID-19 myocarditis subgroups (*p* > 0.05 for all comparisons).

Late Gadolinium Enhancement (LGE) Findings

Among all patients with COVID-19 myocarditis, LGE was observed in 61% (regardless of vaccination status). Of the 17 unvaccinated patients with contrast-enhanced images, 12 (71%) showed myocardial LGE. Among the 6 vaccinated patients with contrast-enhanced images, 2 (33%) demonstrated LGE. Overall, 86% of patients with LGE were unvaccinated.

The most common LGE pattern was subepicardial enhancement, observed in 50% of COVID-19 myocarditis patients (Figure 2), followed by mid-wall (8%) and ischemic (4%) patterns.

In the COVID-19 myocarditis group, the prevalence of LGE and subepicardial enhancement was 61% and 50%, respectively. In comparison, the non-COVID-19 myocarditis group demonstrated LGE in 78% of patients, with subepicardial enhancement in 91%.

There was no statistically significant difference in LGE frequency between the early and late post-infection groups.

## 4. Discussion

This study investigated late-stage myocarditis associated with COVID-19 using CMR, including tissue characterization via T1 and T2 mapping. While previous studies have primarily focused on global T1 and T2 measurements, our study highlights the importance of segmental analysis in identifying localized myocardial involvement.

In addition to the imaging-based findings, we explored the relationship between myocardial tissue characteristics and clinical symptomatology. Although a high proportion of patients reported persistent symptoms such as palpitations and dyspnea beyond four weeks after infection, segmental mapping values were generally similar between symptomatic and asymptomatic patients. However, a significantly higher T1 value was observed in the inferior myocardial segment in symptomatic individuals (1049.7 ± 71.4 ms vs. 984.7 ± 21.8 ms, *p* = 0.024), potentially indicating localized fibrotic changes associated with prolonged symptomatology.

Our findings revealed no significant differences in EF or mapping values between vaccinated and unvaccinated patients. Furthermore, segmental T1 and T2 mapping values were similar between the early (<90 days) and late (>90 days) COVID-19 myocarditis group and healthy controls. However, patients with reduced EF demonstrated significantly increased segmental T1 and T2 values, supporting the association between functional impairment and tissue changes. These results suggest that routine post-COVID CMR imaging in asymptomatic patients may not consistently reveal significant myocardial changes, particularly in those without functional decline. However, the recent literature indicates that myocardial edema—reflected by prolonged T2 relaxation time—remains a common feature, even months after COVID-19 infection, underscoring the persistence of myocardial involvement [11]. Nevertheless, mapping techniques may still offer value in selected cases with reduced EF or prolonged symptoms.

Previous case reports have described subepicardial LGE patterns in acute COVID-19 myocarditis. For instance, a case involving a female patient who developed myocarditis two weeks after confirmed COVID-19 showed subepicardial enhancement in the anterior and anteroseptal myocardial segments along with reduced EF on cardiac MRI [12]. Another case series reported epicardial enhancement in the basal lateral wall and globally increased T1 values during acute COVID-19 myocarditis [13]. Although our study did not include patients in the acute phase, the predominance of subepicardial enhancement was consistent with these earlier findings.

Puntmann et al. conducted a study with 58 patients evaluated approximately 2–3 months post-infection using similar T1 and T2 mapping sequences as in our study. Their results demonstrated significantly elevated T1 values, but not T2 values, compared to healthy controls [1]. In contrast, Ulloa et al. used a 3[3]5 MOLLI sequence in a cohort of 57 patients scanned an average of 81 ± 27 days after infection. They found no significant differences in T1 values but reported significantly elevated T2 values in the COVID-19 group [4]. Similarly, a prospective study of 20 patients with persistent symptoms at 12 weeks post-infection showed no significant difference in global T1 or T2 values between the COVID-19 and control groups, with only one patient showing subepicardial enhancement [14]. In our cohort, although T1 and T2 relaxation times were numerically higher in patients with COVID-19 myocarditis compared to healthy controls, these differences did not reach statistical significance (*p* > 0.05), even after stratifying patients into early and late post-infection subgroups. This finding suggests that post-infectious myocardial tissue alterations may not be consistently detectable using conventional T1 and T2 mapping techniques within the observed time window. The variability in T1 and T2 findings across different studies may reflect heterogeneity in CMR protocols, patient selection, and disease phase at the time of imaging. It is also possible that the sample size in our study was insufficient to detect small-to-moderate differences, underscoring the need for future studies with larger, well-powered cohorts.

Although differences between vaccinated and unvaccinated groups were explored using effect size metrics, the small number of the unvaccinated subgroup (n = 5) limited the statistical power of the comparisons. Moderate effect size was observed in segment 2 T2 mapping (g = 0.62), but post hoc analysis confirmed insufficient power (<0.25) to draw definitive conclusions. Interpretation of vaccination-related results should be approached cautiously due to limited statistical power. Future studies with larger and balanced subgroups are necessary to confirm whether vaccination status influences post-COVID myocardial recovery.

Even though cases of post-vaccine myocarditis were excluded from this study, previous reports have described generally favorable outcomes in these patients, with minimal or no residual myocardial injury on late CMR imaging [11]. In contrast, our COVID-19 myocarditis cohort demonstrated persistent T1 elevation and subepicardial LGE in a considerable proportion of patients, even months after infection. This suggests that while both entities may share similar clinical presentations, their long-term myocardial remodeling patterns could differ significantly.

To the best of our knowledge, this is one of the first studies to present a detailed segmental comparison of T1 and T2 mapping values in COVID-19–related myocarditis, non-COVID-19 myocarditis, and healthy controls. The inclusion of a pre-pandemic myocarditis group allows for a more accurate differentiation between COVID-19–specific myocardial changes and general post-viral myocarditis patterns.

These discrepancies across studies may be attributed to several factors, including variations in underlying cardiac conditions, symptom severity, patient immune response, and differences in mapping protocols and sequence sensitivity. It remains to be clarified whether these imaging findings correlate with clinical outcomes such as arrhythmias, heart failure progression, or long-term functional decline. Prospective studies incorporating myocardial biopsy or serum biomarkers could enhance the understanding of the underlying pathophysiology of COVID-19 myocarditis.

### Limitations

This study has several limitations. First, the sample size in all groups was relatively small, which may affect statistical power and generalizability. Second, patients were not evaluated during the acute phase of COVID-19 infection, potentially missing early myocardial changes. Third, none of the included patients had severe clinical presentations; all had mild to moderate symptoms, which may have limited the range of observed pathology. Additionally, the retrospective single-center design may introduce selection bias. Future longitudinal studies are needed to determine whether the observed segmental abnormalities resolve over time or evolve into chronic myocardial damage.

Multicenter studies including a larger patient population are required to ensure the generalization of the findings. We also believe that providing the necessary conditions and isolation and performing the MRI examinations in the acute period will contribute to the literature in order to have more information about COVID-19 myocarditis.

## 5. Conclusions

This study provides a comprehensive evaluation of late-stage COVID-19-related myocarditis using CMR, including segmental T1 and T2 mapping, comparison with healthy controls, and pre-pandemic myocarditis cases. While no statistically significant differences were observed in mapping values between groups, segmental analyses revealed trends toward localized myocardial involvement, particularly in symptomatic patients and those with reduced ejection fraction. The frequent detection of subepicardial late gadolinium enhancement and modestly elevated T1 values suggests that subtle fibrotic remodeling may persist long after infection. These findings underscore the value of tissue characterization in post-COVID cardiac assessment—especially in patients with clinical symptoms or functional decline. Further large-scale, prospective studies incorporating long-term clinical outcomes, histopathologic validation, and biomarker correlation are warranted to elucidate the pathophysiological underpinnings and prognostic implications of post-COVID myocardial involvement.

## Figures and Tables

**Figure 1 diagnostics-15-02441-f001:**
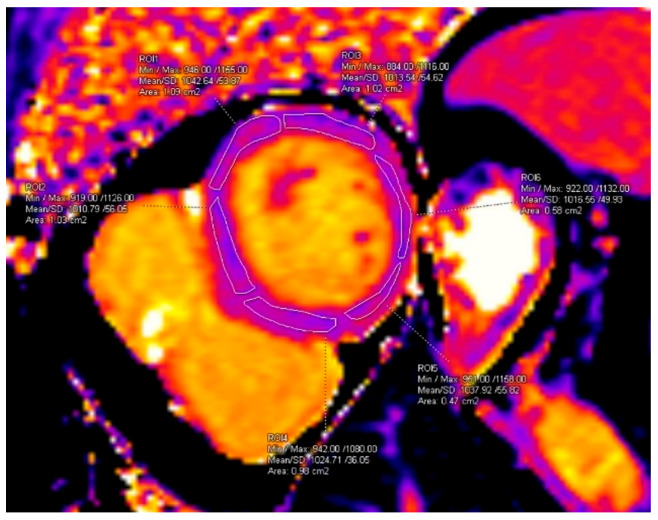
Segmental measurement of T1 values. ROI area values are displayed automatically by the software as “cm2” and correspond to cm^2^.

**Figure 2 diagnostics-15-02441-f002:**
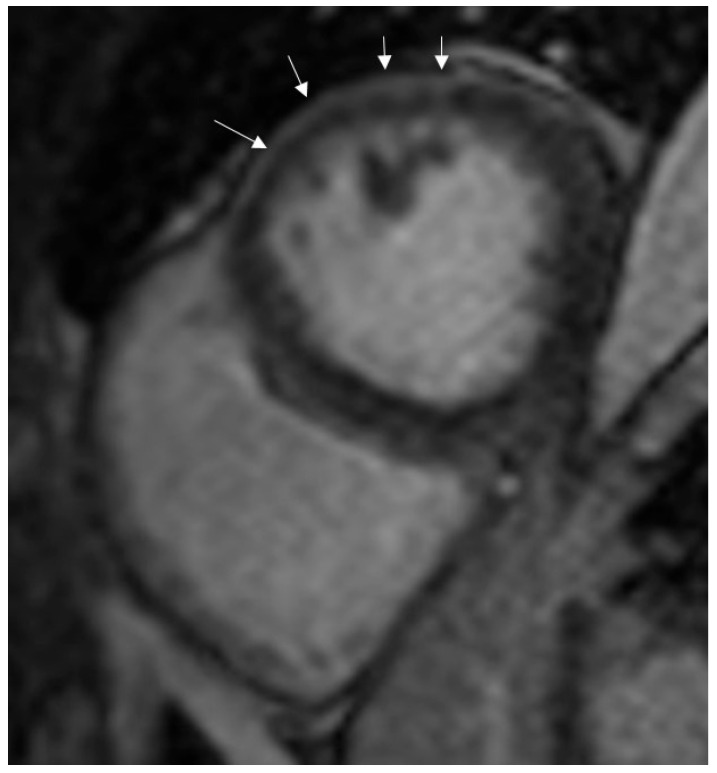
Subepicardial enhancement in late-gadolinium images (arrows).

**Table 1 diagnostics-15-02441-t001:** T1 and T2 mapping values of normal and low ejection fraction groups.

	Normal EF Group	Low EF Group	*p* =
T1 map segment 2	1009.1 ± 37.5	1045.8 ± 37.7	0.023
T1 map segment 5	996.7 ± 31.1	1031.5 ± 31.9	0.025
T2 map segment 5	46.8 ± 2.8	49.7 ± 1.9	0.020
T2 map segment 6	46.3 ± 2.5	48.1 ± 1.8	0.044

EF: Ejection fraction. Statistically significant values are shown. *p* < 0.05 is statistically significant.

**Table 2 diagnostics-15-02441-t002:** Correlation between T1 and T2 mapping values and ejection fraction.

	EF-Spearman’s Rho (*p* = )
T1 map segment 2	−0.549 (0.05)
T1 map segment 1	−0.503 (0.012)
T1 map segment 5	−0.499 (0.013)
T1 map segment 3	−0.452 (0.027)
T2 map segment 6	−0.437 (0.029)

EF: Ejection fraction. Statistically significant values are shown. *p* < 0.05 is statistically significant.

**Table 3 diagnostics-15-02441-t003:** T1 and T2 mapping values of COVID-19 and non-COVID-19 myocarditis groups.

	Non COVID-19 Myocarditis	COVID-19 Myocarditis	*p* =
T1 map segment 1	1030.1 ± 20.6	998.1 ± 37.1	0.034
T1 map segment 4	1090.6 ± 52.7	1014 ± 41	0.001
T1 map segment 5	1069.2 ± 66.7	1006.2 ± 35.3	0.001
T1 map segment 6	1051.6 ± 57.3	995.3 ± 34.5	0.019
T2 map segment 1	48.8 ± 1.9	46.5 ± 1.8	0.002
T2 map segment 4	52.6 ± 3.9	46.7 ± 2.4	0.007
T2 map segment 5	51.7 ± 5.6	47.6 ± 2.9	0.005
T2 map segment 6	50.5 ± 5.5	46.8 ± 2.4	0.008

Statistically significant values are shown. *p* < 0.05 is statistically significant.

## Data Availability

The data presented in this study are available from the corresponding author upon reasonable request.

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
