# Peer review of "Cardiac MRI for COVID-19-Related Late Myocarditis: Functional Parameters and T1 and T2 Mapping"

_diagnostics, 2025, doi:10.3390/diagnostics15192441_

Round 1

Reviewer 1 Report

Comments and Suggestions for Authors

The topic of late-stage cardiac involvement after COVID-19 infection is highly relevant, and your use of cardiac MRI with segmental T1/T2 mapping provides valuable new insights. The manuscript is generally well written and methodologically sound. Below my comments aimed at strengthening your work:

  1. The relatively small number of patients, particularly in the vaccination subgroup, limits the strength of conclusions. Please consider including effect size or post-hoc power calculations, and present vaccination-related findings more cautiously.
  2. The interval between infection and MRI is quite variable (36–565 days). This heterogeneity could influence myocardial changes. Stratifying patients into early vs. late post-infection groups may reveal additional trends and improve interpretation.
  3. The study would be strengthened by providing more detail on patient symptoms and outcomes (e.g., persistence of dyspnea, arrhythmia burden, exercise intolerance). Linking imaging findings with clinical course will highlight the relevance of segmental mapping.
  4. Since T1 values were elevated while T2 values were not, please expand discussion on whether these findings suggest residual fibrosis rather than ongoing inflammation, and how this compares with prior literature.
  5. Figures 1 and 2 would benefit from clearer labeling and higher quality resolution. Some tables (e.g., 1–4) have inconsistent decimal formatting (commas instead of periods). Please standardize.

Author Response

Comments 1: The relatively small number of patients, particularly in the vaccination subgroup, limits the strength of conclusions. Please consider including effect size or post-hoc power calculations, and present vaccination-related findings more cautiously.

Response 1: 

We appreciate the reviewer’s valuable comment regarding the limited sample size, particularly in the vaccination subgroup, and the importance of evaluating effect size and statistical power. In response to this concern, we have implemented the following revisions in the manuscript:

We added this sentence to materials and methods part:

Hedges’ g was used to quantify the effect size between vaccinated and unvaccinated groups due to the small and unequal sample sizes. Additionally, post-hoc power analyses (α = 0.05) were conducted to assess the statistical power of each comparison.

We added this sentence to results:

Effect size analysis revealed small Hedges’ g values for all myocardial segments for both T1 and T2 mapping, with the exception of segment 2 T2 (g = 0.62), indicating a moderate effect. However, due to the small sample size in the unvaccinated subgroup (n=5), post-hoc power remained low (< 0.25) for all comparisons.

We added this sentence to discussion:

Although differences between vaccinated and unvaccinated groups were explored using effect size metrics, the small number of unvaccinated subgroup (n=5) limited the statistical power of the comparisons. Moderate effect size was observed in segment 2 T2 mapping (g = 0.62), but post-hoc analysis confirmed insufficient power (< 0.25) to draw definitive conclusions. Interpretation of vaccination-related results should be approached cautiously due to limited statistical power. Future studies with larger and balanced subgroups are necessary to confirm whether vaccination status influences post-COVID myocardial recovery.

Comments 2: 

  1. The interval between infection and MRI is quite variable (36–565 days). This heterogeneity could influence myocardial changes. Stratifying patients into early vs. late post-infection groups may reveal additional trends and improve interpretation.

Response 2: 

We highly appreciate your suggestion regarding the heterogeneity in the timing between infection and CMR imaging. We found your comment to be very appropriate, and in response, we have revised the manuscript accordingly. Specifically, we stratified the patients into early (≤90 days) and late (>90 days) post-infection subgroups and analyzed T1, T2, and LGE findings across these groups. The relevant details have been added to the Materials and Methods, Results, and Discussion sections;

We added this sentence to material methods section:

To address potential heterogeneity related to timing, patients were stratified into early (≤ 90 days) and late (> 90 days) post-infection subgroups based on the interval between positive PCR and CMR imaging (.)

We added these paragraphs to results section:

Group Comparisons: Early (<90 days) and Late (>90 days) COVID Myocarditis vs. Controls

Although patients were stratified into early (≤ 90 days post-COVID) and late (> 90 days post-COVID) groups based on the interval between infection and MRI, no statistically significant differences were observed between the groups in terms of segmental T1 and T2 values (p > 0.05 for all comparisons). No statistically significant differences in T1 or T2 mapping values were observed between the control group and either the early or late COVID-19 myocarditis subgroups (p > 0.05 for all comparisons).

There was no statistically significant difference in LGE frequency between the early and late post-infection groups.

We added this paragraph to discussion section:

In our cohort, although T1 and T2 relaxation times were numerically higher in patients with COVID-19 myocarditis compared to healthy controls, these differences did not reach statistical significance (p > 0.05), even after stratifying patients into early and late post-infection subgroups. This finding suggests that post-infectious myocardial tissue alterations may not be consistently detectable using conventional T1 and T2 mapping techniques within the observed time window. The variability in T1 and T2 findings across different studies may reflect heterogeneity in CMR protocols, patient selection, and disease phase at the time of imaging. It is also possible that the sample size in our study was insufficient to detect small-to-moderate differences, underscoring the need for future studies with larger, well-powered cohorts.

Comments 3: 

  1. The study would be strengthened by providing more detail on patient symptoms and outcomes (e.g., persistence of dyspnea, arrhythmia burden, exercise intolerance). Linking imaging findings with clinical course will highlight the relevance of segmental mapping.

Response 3: 

We appreciate the reviewer’s insightful suggestion regarding the need to correlate imaging findings with clinical symptoms and outcomes. In response:

We added this sentence to material-methods:

Clinical records were reviewed for symptoms such as dyspnea, chest pain, palpitations, and reduced exercise tolerance. Symptom persistence beyond 4 weeks was considered as a 'prolonged clinical course'.

We added these sentences to results:

The most common presenting symptom was palpitations (n = 12), chest pain (n = 9) and dyspnea (n = 6), followed by syncope (n=1) and reduced excercise tolerance  (n = 1).

Persistent symptoms beyond 4 weeks were reported in 78% (n=14/18) of patients with available medical records, most commonly palpitations (n=11), chest pain (n = 2) and dyspnea (n = 2).

Among the cohort, four patients experienced major adverse cardiac outcomes, which included new-onset heart failure, catheter ablation for arrhythmia, ICD implantation, and a non-fatal myocardial infarction.

When patients with and without persistent symptoms (based on available medical records) were compared, no significant differences were observed in myocardial mapping values, except for the T1 value obtained from the inferior wall, which was significantly higher in the symptomatic group (1049.7 ms vs. 984.7 ms, p = 0.024).

We added this sentence to discussion:

Although a high proportion of patients reported persistent symptoms such as palpitations and dyspnea beyond four weeks after infection, segmental mapping values were generally similar between symptomatic and asymptomatic patients. However, a significantly higher T1 value was observed in the inferior myocardial segment in symptomatic individuals (1049.7±71.4 ms vs. 984.7±21.8 ms, p = 0.024), potentially indicating localized fibrotic changes associated with prolonged symptomatology.

 Comments 4: 

  1. Since T1 values were elevated while T2 values were not, please expand discussion on whether these findings suggest residual fibrosis rather than ongoing inflammation, and how this compares with prior literature.

Response 4: 

We sincerely thank the reviewer for this important comment. We would like to clarify that in the original version of the manuscript, the results were inaccurately phrased, which may have led to the impression that T1 values were elevated while T2 values were not. In fact, both T1 and T2 relaxation times were numerically higher in the COVID-19 myocarditis group compared to controls, but these differences did not reach statistical significance (p > 0.05).

To address this, we have corrected the relevant statements in the Results and Discussion sections. The revised text now reads:

Furthermore, segmental T1 and T2 mapping values were similar between the early (<90 days) and late (>90 days) COVID-19 myocarditis group and healthy controls. However, patients with reduced EF demonstrated significantly increased segmental T1 and T2 values, supporting the association between functional impairment and tissue changes. These results suggest that routine post-COVID CMR imaging in asymptomatic patients may not consistently reveal significant myocardial changes, particularly in those without functional decline. However, recent literature indicates that myocardial edema—reflected by prolonged T2 relaxation time—remains a common feature, even months after COVID-19 infection, underscoring the persistence of myocardial involvement [11]. Nevertheless, mapping techniques may still offer value in selected cases with reduced EF or prolonged symptoms.”

We apologize for this oversight and believe that the revised wording now more accurately reflects our findings.

Comment 5: 

  1. Figures 1 and 2 would benefit from clearer labeling and higher quality resolution. Some tables (e.g., 1–4) have inconsistent decimal formatting (commas instead of periods). Please standardize.

response 5: 

Some tables (e.g., 1–4) have inconsistent decimal formatting (commas instead of periods). Please standardize. à We thank the reviewer for pointing this out. The decimal formatting in all tables has been revised, and commas were replaced with periods to ensure consistency across Tables 1–3.

Figures 1 and 2 would benefit from clearer labeling and higher quality resolution. à We appreciate the reviewer’s helpful suggestion. Figures 1 and 2 have been revised with clearer labeling and improved resolution to enhance readability and presentation quality.

We hope that these additions appropriately address the reviewer’s concern and improve the clarity and transparency of the manuscript.

Reviewer 2 Report

Comments and Suggestions for Authors

This is a well-written manuscript. The comparison of late Covid 19 myocarditis to normal control is good. 

However, comparing the late Covid 19 myocarditis imaging to early (acute) non-covid 19 myocarditis does not seem appropriate. I can understand that the data were available from the authors' previous publication (6). The timing is important. It was mentioned in the methods, in the results (briefly), but not in the discussion.

The segmental comparison is interesting, and relatively new.

I suggest that the authors my compare their Covid 19 myocarditis results with late non-Covid 19 myocarditis results.

It may be beyond the scope of the manuscript, but an interesting comparison can be to post vaccination myocarditis late imaging results.

Author Response

Comment 1: However, comparing the late Covid 19 myocarditis imaging to early (acute) non-covid 19 myocarditis does not seem appropriate. I can understand that the data were available from the authors' previous publication (6). The timing is important. It was mentioned in the methods, in the results (briefly), but not in the discussion.

Response 1: 

We thank the reviewer for this important observation. We fully agree that comparing late-stage COVID-19 myocarditis directly with acute non-COVID myocarditis would not be appropriate due to differences in disease stage. To address this, we also evaluated a subgroup of patients with late non-COVID myocarditis (time interval 22–1815 days, mean 319 ± 597, median 60 days). This allowed us to make a more balanced comparison between two cohorts both imaged in the late phase.

We have now clarified this point in the Discussion section:

Since the COVID-19 myocarditis group predominantly consisted of patients in the late post-infection period, we selected non-COVID myocarditis cases who also underwent late-phase imaging for a more accurate comparison. In the non-COVID myocarditis group, the interval between diagnosis and CMR ranged from 22 to 1815 days (mean: 319 ±â€Ż597 days; median: 60 days). Although the non-COVID group underwent MRI at a numerically later time point compared to the COVID-19 group (319 vs. 164.4 days), this difference was not statistically significant.

Comment 2:It may be beyond the scope of the manuscript, but an interesting comparison can be to post vaccination myocarditis late imaging results.

Response 2: 

We thank the reviewer for this thoughtful suggestion. We agree that comparing our findings with post-vaccination myocarditis on late cardiac MRI would be of interest. However, as acknowledged in the Methods, patients with post-vaccination myocarditis were excluded from our study design, and therefore a direct comparison was not feasible within the scope of the present analysis.

Nevertheless, we have expanded the Discussion to briefly address this point in the context of prior literature. We added this sentece to discussion part:

Even though cases of post-vaccine myocarditis were excluded from this study, previous reports have described generally favorable outcomes in these patients, with minimal or no residual myocardial injury on late CMR imaging [11]. In contrast, our COVID-19 myocarditis cohort demonstrated persistent T1 elevation and subepicardial LGE in a considerable proportion of patients, even months after infection. This suggests that while both entities may share similar clinical presentations, their long-term myocardial remodeling patterns could differ significantly.

Response 2: 

Round 2

Reviewer 1 Report

Comments and Suggestions for Authors

This revision is substantially improved and now provides a clearer, more comprehensive analysis of late-stage COVID-19 myocarditis. The added clinical outcome data and statistical rigor strengthen the manuscript’s scientific value.

Reviewer 2 Report

Comments and Suggestions for Authors

The revised version of this manuscript has been significantly improved.

As the topic is "hot", especially after the recent ESC Immune Myocardial and Pericardial disease 2025 guidelines, and the results showing late gadolinium also in the subepicardium in the majority of patients.